# Improved Catalytic Propylene Epoxidation for Extruded Micrometer TS-1: Introducing Mesopores and Macropores Insides the Crystals

Jiangbo Li, Feifei Zhang, Lukuan Zong, Xiangyu Wang * and Huijuan Wei *

Green Catalysis Center, College of Chemistry, Zhengzhou University, Zhengzhou 450001, China; lijiangbozzu@163.com (J.L.); feifeizhangzzu@163.com (F.Z.); lkzong@gs.zzu.edu.cn (L.Z.)
* Correspondence: wangxiangyu@zzu.edu.cn (X.W.); weihuijuan@zzu.edu.cn (H.W.)

**Abstract:** In the paper, mesopores and macropores are introduced inside the crystals of micrometer microporous titanium silicate-1 (TS-1) to solve the problem of active site coverage and mass transfer during extrusion. Hierarchically porous titanium silicalite-1 (HPTS-1) was acquired by treating micrometer microporous TS-1 with TPABr and ethanolamine. Extruded HPTS-1 maintained greatly superior catalytic performance and possessed high mechanical strength. Characterization results showed that extruded HPTS-1 possessed macroporous, mesoporous structure inside the crystals. These abundant pores are not only beneficial for diffusion reactants, but also make Ti-peroxo species ($\eta^2$), active oxidation sites in TS-1/$H_2O_2$ system become much more reactive. The formula of extruded HPTS-1 was optimized using an orthogonal experiment. The maximum strength of extruded HPTS-1 was up to 200 N/cm, the highest yield of propylene oxide was 92.5% and the specific rate was up to 41.9%. The research provides a scientific basis for producing extruded catalysts with excellent catalytic performance and high mechanical strength in industrial applications.

**Keywords:** mesopores and macropores inside the crystals; extruded catalysts; TS-1; propylene epoxidation





## 1. Introduction

Propylene oxide (PO) is a crucial intermediate of the chemical industry for producing polyurethane plastics, polyether polyols, unsaturated resins, surfactants and other products [1–8]. Traditional processes of producing PO are the chlorohydrin method, co-production method and hydrogen peroxide direct oxidation method (HPPO) [9–14]. In recent years, the HPPO process, based on titanosilicate catalysts (TS-1), has attracted many scholars' attention due to it being green, low-investment and high-efficiency [15–20].

TS-1 was widely used in selective oxidation reaction with hydrogen peroxide, such as partial oxidation of alkanes [21], epoxidation of alkenes [22], hydroxylation of aromatics [23], ammonia oxidation of ketones [24], partial oxidation of alcohol [25] and partial oxidation of organic sulfide [26]. The powder TS-1 applied in industrial production has some defects such as clogging up pipes, difficult separation and other problems. Therefore, shaping is necessary for the powder TS-1 catalyst in order to decrease the pressure drop, achieve a moderate mechanical strength and have a good tolerance to plugging by dust. The current shaping is molded as follows: particles, spheres, extrudates and so on [27,28]. Extrusion molding is a very important shaping technology for producing solid catalysts [29,30]. Large amounts of binder such as silica sol are often added into the extrusion process to make the catalysts possess excellent mechanical strength. While, the addition of binder not only makes the effective composition of catalysts decrease but also blocks some pores and covers active sites. Therefore, the catalytic performance of extruded catalyst is often sharply reduced. It is still a challenge to produce extruded catalysts with excellent catalytic performance and high mechanical strength.



It is reported that adding binders of large particle size can improve the porosity and mass transfer efficiency of the extruded catalyst [31]. Small particle size TS-1 can provide shorter channels and larger external surface and can effectively reduce the binder coverage in the extrusion process. Zuo reported that a small-crystal TS-1 with $600 \times 400 \times 250$ nm was extruded, and the conversion of $H_2O_2$ and the yield of PO could reach 90–95% and 81%, respectively [32]. In addition, introducing the mesopore is also an effective method to reduce the binder coverage in the extrusion process. Wu et al. studied extrusion of hierarchically porous ZSM-5 and the influence of several process factors on the strength and textural properties [33]. Zuo and his co-workers also treated TS-1 extrudates using TPAOH to introduce the mesopore, and the final extrudates reached the strength of 150 N/cm and gained good catalytic performance in propylene epoxidation [34].

In this paper, to solve the problem of active site coverage and mass transfer during extrusion, mesopores and macropores are introduced inside the crystals of micrometer microporous TS-1. TS-1 was prepared by the hydrothermal method using cheap silica sol and tetrabutyl titanate (TBT) in the TPABr-ethanolamine system. Additionally, hierarchically porous titanium silicalite-1 (HPTS-1) was obtained by the modification of TS-1. HPTS-1 and TS-1 were both extruded by mixing with their powder, silica sol and other additives, respectively. An orthogonal experiment was also designed to examine the effect of some factors such as binder (silica sol), forming-pore agent (sesbania powder), water-retaining agent (polyvinyl alcohol, abbreviated as PVA) and lubricant (starch) on the strength of extruded HPTS-1 catalysts and the yield of PO.

## 2. Results

### 2.1. Characterization of Extruded TS-1 and HPTS-1 Samples

Scanning electron microscopy (SEM) and Transmission electron microscopy (TEM) images of TS-1 and HPTS-1 are present in Figure 1. Both ends of TS-1 are relatively smooth, exhibiting round-boat shape and the crystal size around $1.22 \times 0.58 \times 0.22$ μm. HPTS-1 gave coffin morphology because the etching is carried out and the phenomenon of dissolution and recrystallization occurs on its surface during the alkaline treatment of TS-1. The crystal size of HPTS-1 is around $1.46 \times 0.58 \times 0.23$ μm. Figure 1b,d shows that there were many mesopores and macropores inside HPTS-1 crystals. What is more, the size of these meso and macropores is over 30 nm, in agreement with the results of pore size distribution.

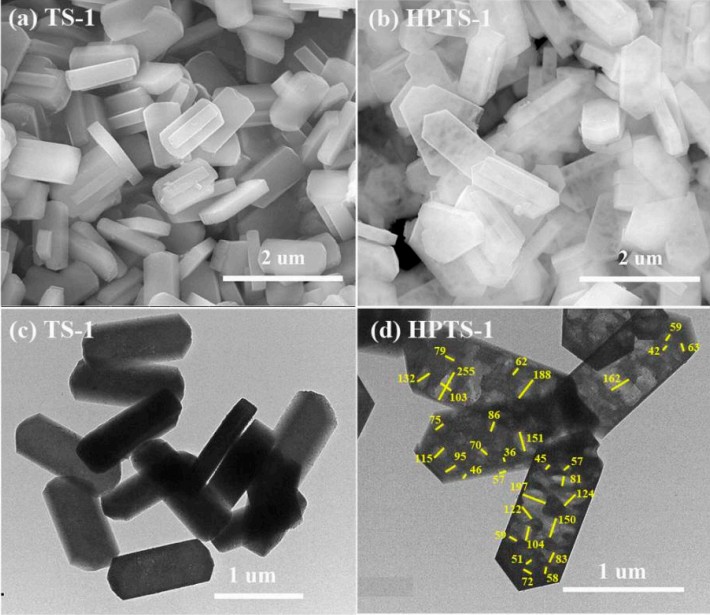

**Figure 1.** SEM (**a**,**b**) and TEM (**c**,**d**) images of TS-1 and hierarchically porous titanium silicalite-1 (HPTS-1) samples.

The powder X-ray diffraction (XRD) patterns of extruded TS-1 and HPTS-1 samples are showed in Figure 2a. All the samples have five characteristic peaks of MFI topology structure located at 7.8°, 8.8°, 23.0°, 23.9° and 24.4° [35–37]. This indicates that the extrusion shaping did not change the framework structure of TS-1 zeolite. First, the relative crystallinity of HPTS-1 decreased because TS-1 was etched by alkaline ethanolamine in the modification process. Then, the relative crystallinity of all extruded catalysts decreased. This is attributed to some amorphous $SiO_2$ introduced by binder during extrusion process. The relative crystallinity of E-TS-1-1 and E-TS-1-2 decreased significantly compared to that of TS-1. However, the relative crystallinity of E-HPTS-1-4 and E-HPTS-1-8 decreased slightly compared to that of HPTS-1, although the extrusion formulas are the same as TS-1 series samples. This indicated that the surface of TS-1 crystal is covered by binder much more intensely for E-TS-1 series (E-TS-1-1 and E-TS-1-2) in the extrusion process.

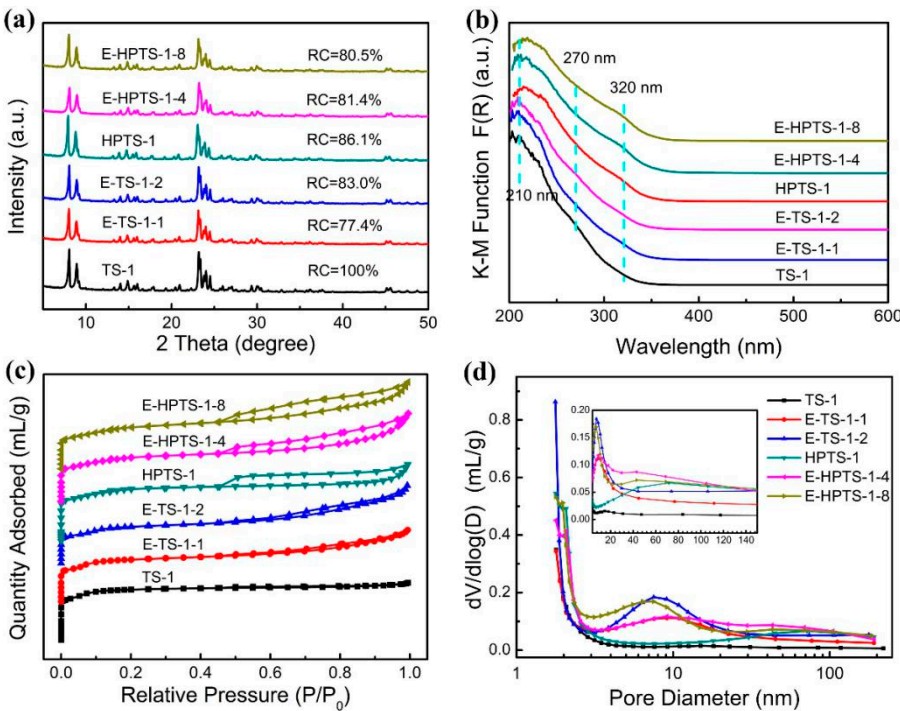

**Figure 2.** (**a**) XRD, (**b**) UV-Vis Diffuse Reflectance Spectra (UV-Vis DRS), (**c**) $N_2$-absorption isotherms and (**d**) pore size distribution of extruded TS-1 and HPTS-1 samples.

UV-Vis spectroscopy has played a key role in the elucidation of the structure of Ti(IV) species in TS-1. All samples have three absorption peaks at 210, 270 and 320 nm, attributed to tetracoordinated framework titanium, hexacoordinated non-framework titanium and anatase $TiO_2$, respectively [32]. The peak around 320 nm of HPTS-1 series samples (HPTS-1, E-HPTS-1-4 and E-HPTS-1-8) is larger than for TS-1 series samples (TS-1, E-TS-1-1 and E-TS-1-2). The reason was that HPTS-1 samples were etched and some framework titanium transferred to anatase $TiO_2$ during the modification process.

$N_2$-absorption isotherms of extruded TS-1 and HPTS-1 samples are demonstrated in Figure 2c. All samples showed remarkable transitions in a low relative pressure ($P/P_0 < 0.2$), showing the microporous structure existing [37]. For TS-1 series samples (TS-1, E-TS-1-1 and E-TS-1-2), with the increase of binder dosage, the hysteresis loops have an increasing trend. This result proved that the formation of mesopores between crystals were introduced in the extrusion process and there was a positive correlation with the content of silica sol. Larger hysteresis loops assigned to the type-IV isotherms were observed at $P/P_0$ from 0.45 to 1 in the series of HPTS-1 samples (HPTS-1, E-HPTS-1-4 and E-HPTS-1-8). This revealed that large amounts of mesopores and macropores are present in the series of HPTS-1 samples. The hysteresis loops of extruded HPTS-1 shrunk

a little because the effective components of extruded HPTS-1 were only 70–80 wt% after the addition of binder. However, with the increase of binder dosage from 20 to 30 wt%, E-HPTS-1-8 has larger hysteresis loops than E-HPTS-1-4.

The pore size distribution of extruded TS-1 and HPTS-1 samples (Figure 2d) is very interesting. When the binder dosage was 20 wt%, the extruded samples (E-TS-1-1 and E-HPTS-1-4) both gave the similar mesopore distribution from 2 to 20 nm. When the binder content increased up to 30 wt%, the extruded samples (E-TS-1-2 and E-HPTS-1-8) also have similar mesopore distribution from 2 to 20 nm. The amount of mesopores from 2 to 20 nm is proportional to the content of silica sol in the extrusion process. It further verified the formation of mesopores between crystals in the extrusion process. For HPTS-1, the mesopores were mainly distributed at over 30 nm. It showed that mesopores and macropores were generated inside the crystals in the modification process.

The formation, upon contact with the $H_2O_2/H_2O$ solution, of a new Ligand-to-Metal Charge Transfer (LMCT) from the peroxidic moiety to the framework titanium appears at around 385 nm, attributed to Ti-peroxo species ($\eta^2$) [38–41]. They are considered the active oxidation sites in the $H_2O_2$-loaded TS-1 sieve [42–46]. Figure 3a,b presents the DRS UV-Vis spectrum of extruded TS-1 and HPTS-1 samples in the $H_2O_2$ and $CH_3OH$ system. The maximum absorption wavelength of TS-1 is located at 381 nm. After extrusion, the wavelength of E-TS-1-1 and E-TS-1-2 gives rise to a blue shift to 377 nm. The reason may be that the framework titanium inside the TS-1 framework is covered by binder much more intensely for the E-TS-1 series (E-TS-1-1 and E-TS-1-2). However, a red shift to 388 nm appeared for HPTS-1. The maximum absorption wavelength for HPTS-1-4 and HPTS-1-8 shifted to 396 and 395 nm, respectively. Red shift of the maximum absorption wavelength indicated that Ti-peroxo species ($\eta^2$) are more reactive. After the introduction of mesopores and macropores inside the crystals, framework titanium in micropores can be exposed and the reactants easily get access to framework Ti species, effectively making Ti-peroxo species ($\eta^2$) are more reactive.

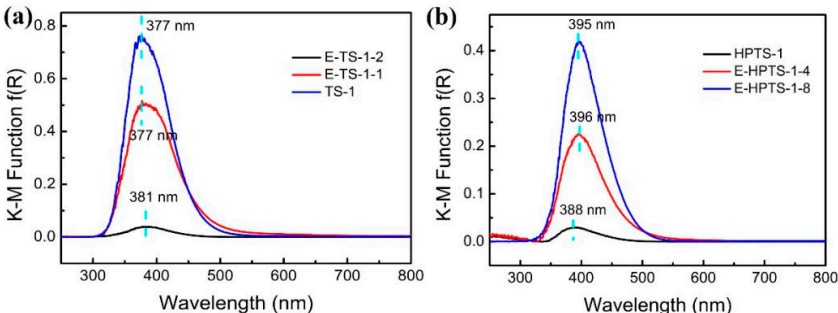

**Figure 3.** DRS UV-Vis obtained from (TS-1/$CH_3OH$/$H_2O_2$−TS-1/$CH_3OH$/$H_2O$) of extruded TS-1 (**a**) and HPTS-1 (**b**) samples.

Table 1 gives the textural properties of TS-1 and HPTS-1 series samples. Compared with TS-1, the micropore area and volume of HPTS-1 obviously decreased, but the external area, total pore and mesopore volume increased markedly. This is due to the large amounts of mesopores and macropores inside the crystals produced in the modification of TS-1. In comparison with TS-1, Brunauer-Emmet-Teller (BET) surface area, micropore area and volume of E-TS-1 series (E-TS-1-1 and E-TS-1-2) fell sharply. The first reason was that the extrusion process only used 70–80 wt% TS-1 samples. The second reason was that many more micropores are blocked by binder for E-TS-1 series (E-TS-1-1 and E-TS-1-2) in the extrusion process. The external surface area, total pore and mesopore volume increased remarkably due to partial mesopores between the crystals introduced in the extrusion process. For the series of HPTS-1 samples (HPTS-1, E-HPTS-1-4 and E-HPTS-1-8), the extrusion process has the same effect on their textural results as for the TS-1 series samples. BET surface area, micropore area and volume of E-HPTS-1 (E-HPTS-1-4 and E-HPTS-1-8)

decreased obviously but the total pore and mesopore volume increased. With the increase in the amount of silica sol in the extrusion process, the trend is much more obvious.

**Table 1.** Textural properties of extruded TS-1 and HPTS-1 samples.

| Samples | $S_{BET}$ | $S_{mic}$ | $S_{ext}$ | $V_{tot}$ | $V_{mic}$ | $V_{meso}$ |
|---|---|---|---|---|---|---|
| | $m^2/g$ | | | $cm^3/g$ | | |
| TS-1 | 419 | 335 | 83.9 | 0.197 | 0.138 | 0.059 |
| E-TS-1-1 | 366 | 258 | 108 | 0.252 | 0.107 | 0.145 |
| E-TS-1-2 | 316 | 166 | 150 | 0.273 | 0.071 | 0.202 |
| HPTS-1 | 416 | 229 | 186 | 0.262 | 0.097 | 0.164 |
| E-HPTS-1-4 | 366 | 178 | 189 | 0.306 | 0.076 | 0.230 |
| E-HPTS-1-8 | 345 | 134 | 211 | 0.302 | 0.059 | 0.243 |

## 2.2. Catalytic Performance of Extruded TS-1 and HPTS-1 Samples in the Batch Reactor

Catalytic performance in propylene epoxidation for TS-1 and HPTS-1 samples have been researched and the detail results are shown in Table 2. First, after TS-1 was extruded, $H_2O_2$ conversion and PO yield fell sharply, and specific rate decreased from 241 to 214 $h^{-1}$ with the increase of binder content. One reason is that many micropores are blocked by binder so that the reactant cannot get access to the active site. Although the extrusion process can generate many mesopores, these mesopores occurred between TS-1 crystals, the reactants still had difficulty touching the active site inside the TS-1 crystals. The other reason is the blue shift of Ti-peroxo species ($\eta^2$), the active sites in the propylene oxidation. E-HPTS-1 samples (E-HPTS-1-4 and E-HPTS-1-8) gave fairly good catalytic performance and their specific rates increased from 248 to 352 $h^{-1}$, this is equivalent to an increase of up to 41.9%. It is due to the fact that many mesopores and macropores inside HPTS-1 crystals improve the openness and reactivity of Ti-peroxo species ($\eta^2$). In other words, these results indicate that binder has a weaker influence on HPTS-1 than TS-1 in extrusion process.

**Table 2.** Catalytic performance of extruded TS-1 and HPTS-1 samples in propylene epoxidation.

| Samples | Binder | $X(H_2O_2)$ | $Y(PO)$ | $S(PO)$ | $U(H_2O_2)$ | Strength | Specific Rate |
|---|---|---|---|---|---|---|---|
| | | % | | | | (N/cm) | $(h^{-1})$ |
| TS-1 | 0 | 97.9 | 88.9 | 99.5 | 91.2 | – | 241 |
| E-TS-1-1 | 20 | 78.9 | 78.3 | 99.5 | 99.7 | 82.1 | 243 |
| E-TS-1-2 | 30 | 48.7 | 47.7 | 99.8 | 98.0 | 190 | 214 |
| HPTS-1 | 0 | 99.7 | 98.4 | 99.0 | 99.7 | – | 248 |
| E-HPTS-1-4 | 20 | 99.7 | 85.6 | 93.0 | 92.3 | 135 | 310 |
| E-HPTS-1-8 | 30 | 99.1 | 92.5 | 98.7 | 94.5 | 200 | 352 |

Reaction condition: 0.24 g catalysts, $n(C_3H_6)/n(H_2O_2)$ = 1.95, 24 mL $CH_3OH$, 3 mL 27.5 wt% $H_2O_2$, reaction time = 1 h.

## 2.3. Orthogonal Experiment Analysis of Extruded HPTS-1 Samples

Characterization and catalytic performance of extruded HPTS-1 in propylene epoxidation are demonstrated in the supporting information (Figures S2–S5 and Tables S1 and S2). Table 3 gives orthogonal analysis results of extruded HPTS-1 samples. In the range analysis, the strength of extruded HPTS-1 samples and the yield of PO are used as evaluation indexes. $K_{ij}$ represents the sum of strength of level i (i = A, B, C, D) for factor j (j = 1, 2, 3) and $\overline{K}_{ij}$ is the average value of $K_{ij}$. $L_{ij}$ represents the sum of the yield of PO of level i (i = A, B, C, D) for factor j (j = 1, 2, 3) and $\overline{L}_{ij}$ is the average value of $L_{ij}$. $R_{1j}$ (or $R_{2j}$) is a range representing the maximum of $\overline{K}_{ij}$ (or $\overline{L}_{ij}$) minus the minimum of $\overline{K}_{ij}$ (or $\overline{L}_{ij}$). The range is used to evaluate the importance of indexes (the strength of extruded HPTS-1 and the yield of PO). Additionally, its value is the larger, the greater of the influence of the factor (A, B, C, D) corresponding to this range. Taking factor B as an example, the calculation formulas are as follows:

**Table 3.** Orthogonal experiment array $L_9$ $(3^4)$.

| Samples | Binder (wt%) | Sesbania Powder (wt%) | PVA (wt%) | Starch (wt%) | Strength (N/cm) | $Y_{PO}$ (%) | Specific Rate $(h^{-1})$ |
|---|---|---|---|---|---|---|---|
| | A | B | C | D | E | F | |
| E-HPTS-1-1 | 10 | 1 | 0 | 1 | 33.5 | 86.7 | 275 |
| E-HPTS-1-2 | 10 | 3 | 5 | 3 | 42.7 | 90.4 | 275 |
| E-HPTS-1-3 | 10 | 5 | 10 | 5 | 61.2 | 89.5 | 275 |
| E-HPTS-1-4 | 20 | 1 | 5 | 5 | 135 | 85.6 | 310 |
| E-HPTS-1-5 | 20 | 3 | 10 | 1 | 151 | 89.2 | 310 |
| E-HPTS-1-6 | 20 | 5 | 0 | 3 | 143 | 88.6 | 310 |
| E-HPTS-1-7 | 30 | 1 | 10 | 3 | 186 | 90.6 | 354 |
| E-HPTS-1-8 | 30 | 3 | 0 | 5 | 200 | 92.5 | 352 |
| E-HPTS-1-9 | 30 | 5 | 5 | 1 | 171 | 86.6 | 340 |
| $K_1$ | 137 | 355 | 377 | 356 | | | |
| $K_2$ | 429 | 394 | 349 | 372 | | | |
| $K_3$ | 557 | 375 | 398 | 396 | | | |
| $L_1$ | 267 | 263 | 268 | 263 | | | |
| $L_2$ | 263 | 272 | 263 | 270 | | | |
| $L_3$ | 270 | 265 | 269 | 268 | | | |
| $R_1$ | 140 | 13 | 16.3 | 13.3 | | | |
| $R_2$ | 2.3 | 3.0 | 2.0 | 2.33 | | | |

$$K_{B1} = E_{B1} + E_{B4} + E_{B7}$$
$$K_{B2} = E_{B2} + E_{B5} + E_{B9}$$
$$K_{B3} = E_{B3} + E_{B6} + E_{B8}$$
$$L_{B1} = F_{B1} + F_{B4} + F_{B7}$$
$$L_{B2} = F_{B2} + F_{B5} + F_{B9}$$
$$L_{B3} = F_{B3} + F_{B6} + F_{B8}$$
$$\overline{K}_{Bj} = K_{Bj}/3 (j = 1, 2, 3)$$
$$\overline{L}_{Bj} = L_{Bj}/3 (j = 1, 2, 3)$$
$$R_{B1} = \max(\overline{K}_{Bj}) - \min(\overline{K}_{Bj}) (j = 1, 2, 3)$$
$$R_{B1} = \max(\overline{L}_{Bj}) - \min(\overline{L}_{Bj}) (j = 1, 2, 3)$$

As seen in Table 3, range analysis gives the order of $R_1$ as $R_{A1} > R_{C1} > R_{D1} > R_{B1}$. This indicates that the binder (factor A) has the most remarkable influence on the strength of extruded HPTS-1, followed by PVA. With the increase of binder content, the strength of E-HPTS-1 samples is significantly enhanced. The value of $R_2$ for the four factors is similar and it proves that these four factors have similar effects on PO yield. Furthermore, the specific rate was similar when the extruded HPTS-1 catalysts had the same binder content in Table 3. It is shown that PO yield is directly proportional to the effective amount of HPTS-1 in extruded catalysts.

The above results indirectly reflect that after introducing meso and macropores inside the micrometer TS-1, the factor in the extrusion process, especially the binder content, has little influence on its PO yield, and E-HPTS-1 can still maintain comparatively high catalytic performance. Therefore, the problem of catalytic degradation caused by active site coverage in the extrusion process can be solved.

### 2.4. Stability and Regeneration Performance of Extruded HPTS-1 Samples

The catalytic stability of E-HPTS-1-8 has been investigated in the fixed bed reactor (Figure 4). At the initial stage of the reaction, fewer than 50 h, the conversion of $H_2O_2$ and yield of PO increased gradually. From 50 to 220 h, the catalytic performance is basically stable.

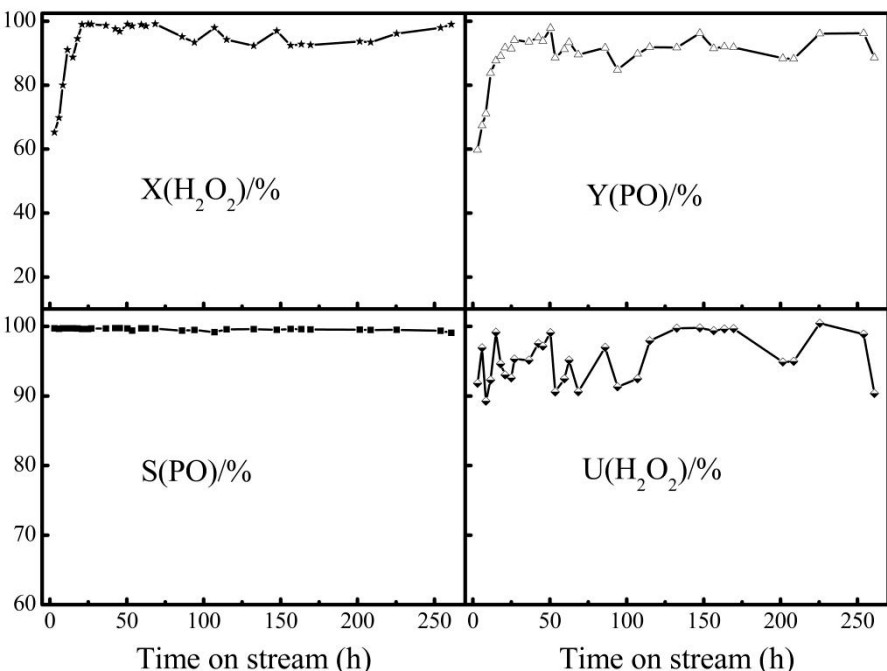

**Figure 4.** The stability of catalytic performance for E-HPTS-1-8 catalyst in fixed-bed reactor. Reaction conditions: reaction temperature = 42 °C; pH of reaction solution = 9.10; $n(C_3H_6)/n(H_2O_2)$ = 2.1; WHSV of propene = 0.64 $h^{-1}$.

The conversion of $H_2O_2$ is above 92% and PO yield is above 90%. When the reaction time is above 250 h, the yield of PO began to fall. The selectivity of PO was generally higher than 99% for the entire run time. In general, E-HPTS-1-8 catalyst still maintained a relatively excellent catalytic performance, even better than the activity shown in the batch reaction. E-HPTS-1-8 after 250 h reaction was regenerated by simple calcination, and then the catalytic performance in batch reactor was investigated (Table 4). The results showed that the catalytic performance of E-HPTS-1-8 was basically recovered after regeneration, the yield of PO and utilization efficiency of $H_2O_2$ are more than those of fresh samples.

**Table 4.** Catalytic performance of fresh and regenerated E-HPTS-1-8 in batch reactor.

| Samples | $X(H_2O_2)$/% | $Y(PO)$/% | $S(PO)$/% | $U(H_2O_2)$/% |
|---|---|---|---|---|
| E-HPTS-1-8 | 99.14 | 92.49 | 98.72 | 94.51 |
| Re-E-HPTS-1-8 | 99.60 | 96.44 | 98.95 | 97.86 |

Reaction condition: 0.24 g catalysts, $n(C_3H_6)/n(H_2O_2)$ = 1.95, 24 mL $CH_3OH$, 3 mL 27.5 wt% $H_2O_2$, reaction time = 1 h.

## 3. Conclusions

In summary, after the mesopores and macropores inside the crystals were introduced by treating microporous TS-1 with TPABr and ethanolamine, the effect of the extrusion process on textural properties is weakened remarkably. Specifically, the decrease trend of the relative crystallinity, total surface area, micropore area and volume significantly reduce after HPTS-1 samples were extruded. There is a superior catalytic performance for the extruded hierarchically porous TS-1 than extruded TS-1 in propylene epoxidation. An orthogonal experiment showed that during the extrusion process, the binder has the largest influence on the strength of extruded HPTS-1 samples. The PO yield is proportional to the effective amount of HPTS-1 in extruded catalysts, regardless of other factors in the extrusion process. In the batch reactor, the conversion of $H_2O_2$ for extruded HPTS-1 catalyst is over 99% and the yield of PO is more than 92.5% for E-HPTS-1-8. Furthermore, the E-HPTS-1-8 sample has always maintained good catalytic stability for 250 h. After

regeneration by simple calcination, the yields of PO and utilization efficiency of $H_2O_2$ are more than those of fresh samples. This paper provides a method to produce extruded catalysts with excellent catalytic performance and high mechanical strength.

**Supplementary Materials:** The following are available online at https://www.mdpi.com/2073-4344/11/1/113/s1, TEM of extruded TS-1 and HPTS-1 (Figure S1), Characterization of extruded HPTS-1 (Figures S2–S5, Table S1) and catalytic performance of extruded HPTS-1 (Table S2).

**Author Contributions:** J.L. and F.Z. designed and performed the experiments; X.W. and H.W. analyzed the data and wrote the paper. L.Z. revised the paper. All authors have read and agreed to the published version of the manuscript.

**Funding:** The Innovation Fund for Elitists of Henan Province, China (Grant No. 0221001200), the Natural Science Foundation of China (No. 21773215) and the Joint Project of Zhengzhou University.

**Institutional Review Board Statement:** Not applicable.

**Informed Consent Statement:** Not applicable.

**Data Availability Statement:** Data is contained within the article.

**Acknowledgments:** Thanks to Li Baojun (Zhengzhou University) and Liu Meng (Shangqiu Normal University) for their support to this work.

**Conflicts of Interest:** The authors declare no conflict of interest.

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
