# Peer review of "Improved Catalytic Propylene Epoxidation for Extruded Micrometer TS-1: Introducing Mesopores and Macropores Insides the Crystals"

_catalysts, doi:10.3390/catal11010113_

Round 1
Reviewer 1 Report
Referee Report on Manuscript Number: ID: catalysts-1060273 Title: Improved catalytic propylene epoxidation for extruded micrometer TS-1: introducing meso and macro pores insides the crystals, by Li et al.
The paper presents an extensive study dealing with the investigation of performance enhancement during the application of micrometer sized TS-1 in catalytic propylene epoxidation as a function of the incorporation of meso- and macropores. As far I am able to correctly judge about the sample preparation and modification, anything appears to be well described so that this paper, offering a wealth of experimental data, might be of some interest for the specialists, notably since the relevance of the binder is thoroughly investigated. For me, it was somehow disappointing that I did not become aware of any efforts with modelling and quantitation by which the origin of the respective improvements would have become more clearly visible. Parallel diffusion studies, such as considered by e.g. L. Gueudré, M. Milina, S. Mitchell, and J. Pérez-Ramírez in “Superior Mass Transfer Properties of Technical Zeolite Bodies with Hierarchical Porosity”, Adv. Funct. Mater. 24 (2014) 209–219 might have helped to get separate information up to which extent improvement of the catalytic properties might be referred to transport enhancement by the incorporation meso/macro (i.e. transport) pores. I am, therefore, indifferent whether the paper in its present stage is already ready for being published in your journal and would like to leave the final decision to really those who are more familiar with the reaction under study and the benefit of the novel results attained in the present study.
Author Response
Your question is very professional. The quantitation relationship between pore diffusion model and catalytic performance needs to be further studied indeed. That's what we lack. And we'll go into that in the future.
Please allow me to make an explanation here. The catalytic propylene epoxidation of extruded micrometer TS-1 is improved by introducing meso and macro pores insides the crystals in this manuscript. In fact, the improvement of catalytic properties are not only based on the diffusion of substrates, but also the chemical properties including the nature of Ti species in the catalyst. Therefore, we think that these abundant pores not only are beneficial for the diffusion reactants, but also make Ti-peroxo species (η2), active oxidation sites in TS -1/H2O2 system, become much more reactive.
Thank you again for providing us with good suggestion.
Reviewer 2 Report
The manuscriot is written very well and the analyses are perfect.
Author Response
Thanks for your positive evaluation gratefully.
Reviewer 3 Report
- Organization of the paper
TS-1 was first reported by Taramasso et al. in 1983 (US Patent N° 4410501). The epoxidation of propene to produce propene oxide was widely studied and commercialized (See the review of F. Cavani et al., ChemSusChem, 2009, 2, 508). Why did the authors use 5 pages of 11 to describe the synthesis, characterization, and textural properties of TS-1 and HPTS-1? This could be added in the supporting information. Table 2 (catalytic performance) and references could be enough.
- Selection of the binder.
The role of the inorganic binders is to provide the extrudate with satisfactory physical strength. However, binders this component may interfere with the catalytic properties of the final extruded particles since additional species and phases are introduced to the composition of the final catalytic system. Consequently, a study of the non-efficient decomposition of hydrogen peroxide could be helpful. Why silica sol, sesbania powder was used? Why did the authors add starch and PVA? No characterization of the products is given by the authors, no commercial references, no information (MW of PVA, % of hydrolysis, …)
- Catalytic performance
Furthermore, could the authors give the amount in mmol of reactant (Section 4.3). What is the ratio of H2O2 /propylene? Did this ratio has an impact on yield and selectivity?
Did the authors study the recycling of their catalyst?
- I don't feel qualified to judge the English language and style. However some sentences are not “chemically” correct: line 158, 259 among others
In conclusion, the paper presents interesting results but should be strongly reorganized and the catalytic evaluation should be strengthened.
Reviewer 4 Report
The authors report on the investigation of the propylene oxidation over TS-1 with meso- and macropores. The present manuscript seems to show proper experimental data and to discuss about the data. However, there are still unverified things that the authors should reconsider.
Some specific comments are provided below for the authors to reconsider.
- A space should not be inserted to between the number and unit. For example, “92.5 %” should be corrected to “92.5%”.
- The figures of Figures S2, S3, and S4 are appearing. They should be double-checked.
- Introduction: The PO production over catalysts has been widely researched by a lot of researchers. Au catalysts have been studied for the PO production using propene. In particular, Haruta reported a lot of researches about the PO production using Au catalysts including Au/TS-1 in gas and liquid phases. It is necessary to introduce these previous researches in the manuscript.
- Introduction: What is “HPTS-1”? When abbreviations are used, formal or original nomenclatures should be shown at first.
- Introduction: Before the objectives are shown in the present manuscript, it is necessary to explain subjects which should be solved. Otherwise, it is very difficult to tell whether the objectives in this study contain some novelties or not in the research field.
- Figure 1: In this study, extruded samples were focused on. Thus, the SEM and TME images of extruded samples are necessary to be shown in the main body.
- Figure 2: Why do not materials used for the extrusio of TS-1 appear in the XRD patterns?
- UV-vis: The authors demonstrate that HPTS-1 contained larger amount of anatase than TS-1 did. Anatase and octahedrally-coordinated Ti species generally active for the decomposition of H2O2 compared with tetrahedrally-coordinated Ti species in TS-1. Thus, differences between the catalysts in Figure 3 are difficult to be explained only based on “Ti species” in the catalysts. Furthermore, the blue shift was observed in TS-1 series, while the red shift was observed in HPTS-1 series in Figure 3. In Figure 2, there are differences in ratios of Ti species between the catalysts; however, critical differences for changing the peak shift are not shown. The authors should interpret the experimental data and discuss them carefully.
- Table 2: What is “Specific Rate”? How did the authors evaluate them?
- Table 3: According to the explanation in Line 171, “K1B” is wrong, and “KB1” is right. Furthermore, what is “factor j (j = 1, 2, 3)”? For example, what are E1, E4, and E7 of “K1B = E1 + E4 + E7”?
- Table 3: The authors should carefully discuss what the experimental data (each value) in Table 3 mean. Furthermore, the authors should discuss what and how factor(s) influence to what species in the reaction system.
- The catalytic properties are discussed based on only the diffusion of substrates and porous properties. However, chemical properties including the nature of Ti species in the catalyst should also influence the catalytic properties. Therefore, the authors should carefully discuss the relations between the chemical properties and the catalytic properties.
- Synthesis of TS-1: It is not necessary to show “n” prior to materials such as SiO2 and TiO2.
- Table 4: What does “%” mean? “mol%” or “wt%”?
- There are some errors of the description and English grammar throughout the present manuscript. The authors should carefully double-check the manuscript.
Round 2
Reviewer 1 Report
I appreciate and understand the authors' repsonse to my concern offered in my first report and would not like to block acceptance of their paper and eventual publication
Author Response
Thank you very much.
Reviewer 3 Report
The new version of the manuscript has been reduced and densified, which is good.
1/ However, there are some editing problems:
Line 68: 2.1 Characterization of extruded TS-1 and HPTS-1 samples
Line 155: 2.1.2 Catalytic performance of TS-1 samples
Line 171: 2.2 Orthogonal experiment results of extruded HPTS-1 samples
Why this numbering? Why is extrusion development last and why it is not discussed in relation to catalytic property?
2/ I appreciate the effort made by the authors to provide characterizations of the products used in the preparation of the extrudate. However (and again), why Sesbania powder? Where come from the idea to use it to improve viscosity and for adhesion and lubrication (references are needed).
3/ I also appreciate the details provided by the authors in their answer, but experimental details must be added for the readers in the caption of Table 2!
4/ In their answer, the authors show a new figure, figure A with the Catalytic performance of E-HPTS-1-8 in a fixed-bed reactor and Table A. Why these important data are not provided in the manuscript?
Reviewer 4 Report
It seems that the authors have responded to reviewer’s comments and revised their manuscript according to reviewer’s comments.
Author Response
Thank you gratefully.
Round 3
Reviewer 3 Report
Corrections have been made. The manuscript could be published.